# Identification of physiological adverse events using continuous vital signs monitoring during paediatric critical care transport: A novel data-driven approach

Milan Kapur[1]*, Kezhi Li[2], Alexander Brown[3], Zhiqiang Huo[4,5], Philip Knight[6], Gwyneth Davies[1,7]‡, Padmanabhan Ramnarayan[6,8,9]‡

1 Department of Population, Policy and Practice, University College London Great Ormond Street Institute of Child Health, London, United Kingdom, 2 Institute of Health Informatics, University College London, London, United Kingdom, 3 Department of Infection, Immunity and Inflammation, University College London Great Ormond Street Institute of Child Health, London, United Kingdom, 4 Wolfson Institute of Population Health, Queen Mary University of London, London, United Kingdom, 5 Department of Population Health Sciences, King's College London, London, United Kingdom, 6 Children's Acute Transport Service, Great Ormond Street Hospital for Children NHS Foundation Trust, London, United Kingdom, 7 Department of Respiratory Medicine, Great Ormond Street Hospital for Children NHS Foundation Trust, London, United Kingdom, 8 Department of Surgery and Cancer, Imperial College London, London, United Kingdom, 9 Department of Paediatric Critical Care, St Mary's Hospital, Imperial College NHS Healthcare Trust, London, United Kingdom

‡ GD and PR are Joint senior authors on this work.
☯ Padmanabhan Ramnarayan and Gwyneth Davies made equal contribution on this work.
* m.kapur@ucl.ac.uk

## Abstract

Interhospital transport of critically unwell children exacerbates physiological stress, increasing the risk of deterioration during transport. Due to the nature of illness and interventions occurring in this cohort, defining "normal" vital sign ranges is impossible, which can make identifying deterioration events difficult. A novel data-driven approach was developed to identify adverse respiratory and cardiovascular events in critically ill children during interhospital transport. In this retrospective cohort study of 1,519 transports (July 2016 to May 2021), vital signs were recorded at one-second intervals and then analysed using an adaptation of Bollinger Bands, a technique borrowed from financial market analysis. This method dynamically established each patient's stable ranges for heart rate, blood pressure, oxygen saturation, and other respiratory parameters, and flagged adverse events when multiple parameters simultaneously fell outside their expected ranges. Adverse respiratory events were identified when oxygen saturation deviated below a dynamically defined threshold alongside at least one additional respiratory parameter. Cardiovascular events were defined by concurrent deviations in blood pressure and heart rate. Overall, 15.6 percent of transports had one or more adverse respiratory events, and 21.5 percent had at least one adverse cardiovascular event. To validate these labels, the number of adverse events and the cumulative duration of vital sign instability during transport

which permits unrestricted use, distribution, and reproduction in any medium, provided the original author and source are credited.

**Data availability statement:** The patient physiological data underlying the findings of this study are third-party data subject to ethical and legal restrictions due to their sensitive patient information content. This dataset was obtained from routine clinical records held by the Children's Acute Transport Service (CATS) and Great Ormond Street Hospital (GOSH). Our research use of this dataset was approved by GOSH Research and Innovation Division, under ethical approvals for use of routine de-identified healthcare and operational hospital data (Research Database, NHS REC reference 21/LO/0646). The authors did not receive any special privileges in accessing the data that other researchers would not have. Researchers seeking access to this dataset for secondary analysis must obtain permission directly from the data custodians. Interested researchers should contact the Research and Innovation Division at Great Ormond Street Hospital for Children NHS Foundation Trust (Research.Governance@gosh.nhs.uk). Data access applications are subject to a formal review process, including confirmation of appropriate ethical approval and the establishment of a Data Sharing Agreement with the requesting institution.

**Funding:** MK is supported by an NIHR Academic Clinical Fellowship at University College London. KL is supported by UKRI Centre for Doctoral Training in AI-enabled healthcare systems. GD is supported by a UKRI Future Leaders Fellowship [MR/T041285/1]. PR is in receipt of grant support from the National Institute of Health Research, Rosetrees Trust and BMA Foundation. The views expressed are those of the authors and not necessarily those of the NHS, the NIHR or the Department of Health and Social Care. The funders had no role in study design, data collection and analysis, decision to publish, or preparation of the manuscript.

**Competing interests:** I have read the journal's policy and the authors of this manuscript have the following competing interests: GD reports speaker honoraria from Vertex Pharmaceuticals and Chiesi Ltd, and advisory board and clinical trial leadership roles with Vertex, unrelated to the current manuscript. PR reports travel support for conference attendance from Fisher and Paykel Healthcare Limited, unrelated to this manuscript. All other authors declare no conflicts of interest.

were compared against clinical markers of deterioration. Each additional respiratory event was associated with increased odds of receiving respiratory support during transport and higher 30-day mortality, while each additional cardiovascular event was associated with increased odds of receiving vasoactive support during transport. Our method detects respiratory and cardiovascular adverse events during transport. The approach is readily adaptable to other high-resolution intensive care datasets, for both retrospective labelling as well as automated, real-time identification of adverse events in the clinical setting, offering a foundation for improved monitoring and early intervention in critically ill patients.

## Author summary

Transporting critically ill children between hospitals is challenging because their condition can worsen during the journey. Previously, studies have relied on fixed vital-sign cutoffs to identify "adverse events," but these rigid thresholds may not account for individual differences in heart rate, blood pressure, and oxygen baseline levels. Here, we used a tool borrowed from financial market analysis to track minute-by-minute vital signs in over 1,500 transport episodes. Our approach identified sudden, patient-specific changes that signalled respiratory or cardiovascular problems. We found that around 15 percent of transfers had at least one breathing-related event and over 20 percent had a heart-related event. Moreover, these events were linked to worse outcomes, such as the need for extra breathing or blood-pressure support. By considering each child's own "stable" vital sign range, this method can detect meaningful changes that standard fixed thresholds might miss. This personalised tracking of vital signs opens the door for further research on routinely collected vital-sign datasets. It could also be deployed in the clinical environment helping clinicians detect a child's deterioration earlier and respond in real time. Although we focused on children in transport, our approach could be adapted to other intensive care settings to improve patient monitoring.

## Introduction

Each year in the United Kingdom, approximately 20,000 children are admitted to Paediatric Intensive Care Units (PICUs) [1]. Many first present to district general hospitals without on-site PICU facilities, necessitating urgent interhospital transfer. Over 4,000 such interhospital transfers were recorded in 2023 [1].

This transfer process can exacerbate physiological stress in critically ill children, increasing the risk of deterioration during transport [2]. Existing research varies substantially in the types of transport-related adverse events studied. A recent systematic review of 40 studies involving over 4,000 intra-hospital paediatric transport episodes identified four main categories of transport-related adverse events: respiratory (e.g.,

hypoxaemia), cardiovascular (e.g., hypotension, tachycardia/bradycardia), equipment-related (e.g., monitor failure), and other (e.g., medication error) [2]. Regarding detection of respiratory and cardiovascular events during transport, most studies focus on quantifying the incidence of certain adverse events or deranged vital-signs [3,4]. It is critical to identify these adverse events as promptly as possible as earlier identification enables earlier intervention potentially leading to improved patient outcomes.

The use of modern electronic healthcare records (EHRs) facilitates the integration and storage of continuous vital signs data captured during paediatric critical care transport [5]. These datasets have the potential to be used for the further study of transport related adverse events enabling us to better understand incidence and risk factors. Furthermore, they may even be used to train AI models, capable of predicting acute risk of deterioration during transport in real-time, enabling clinicians to risk stratify and intervene in a more timely manner [6]. However, a major obstacle exists: routine monitoring data often lack explicit, timestamped labels marking adverse events [5,7].

Despite advances in monitoring during paediatric critical care transport, retrospective labelling of transport-related adverse events from vital sign data remains unexplored. While well-characterized normal vital sign ranges exist for stable paediatric patients, there are no "normal" ranges for critically unwell children during transport; the expected values vary significantly depending on age, diagnosis, severity of illness and ongoing interventions and individualised treatment targets [5,8–11]. This makes labelling datasets for adverse events difficult.

In this study, we propose and validate a novel data-driven approach for automatically labelling patient-related adverse events, specifically respiratory and cardiovascular, using routinely collected vital signs data from the transport of critically unwell children. Our approach addresses the challenge of identifying adverse events in the absence of fixed thresholds by leveraging personalised vital sign trajectories to detect both the occurrence and timing of events. Crucially, we assess the clinical meaningfulness of these identified events by investigating their association with key patient outcomes, namely the receipt of respiratory and vasoactive support during transport, and 30-day mortality.

## Methods

### Study approval

The study was approved by Great Ormond Street Hospital's (GOSH) Research and Innovation Department, under ethical approvals for use of routine de-identified healthcare and operational hospital data (Research Database, NHS REC reference 21/LO/0646).

### Data sources

This retrospective study analysed continuously monitored vital signs of critically ill children transported by the Children's Acute Transport Service (CATS), a regional paediatric critical care team in North London, UK. Since 2016, CATS has used SwiftCare (Kinseed Limited, UK) to collect high-resolution (one data point per second) vital signs, including heart rate (3-lead ECG), blood pressure (invasive and non-invasive), oxygen saturation ($SpO_2$), end-tidal carbon dioxide ($EtCO_2$) and respiratory rate (impedance pneumography and $EtCO_2$ derived). Non-invasive blood pressure was measured using a blood pressure cuff intermittently, and unlike the other parameters (which provided one data point per second), its last measured value was carried forward between readings. Data collection starts at transfer initiation, with ambulance staff using a SwiftCare-enabled smartphone to connect wirelessly to a Philips Intellivue MP5 monitor, recording continuously until patient handover. Linked de-identified electronic health records (EHR) provided additional data, including age, weight, primary diagnosis, transfer respiratory and vasoactive support levels, and 30-day mortality.

While most transport episodes represent unique patients, it is possible that a small proportion of patients were transported on more than one occasion. Due to the anonymisation of records, we were unable to identify repeated transports for the same patient and therefore each episode was treated as a unique record as each transport was distinct.

## Inclusion criteria

Episodes were included if the patient was ≤ 18 years old and had at least 30 minutes of vital signs data during transport recorded. Continuous data throughout transport was not mandated, acknowledging real-world interruptions such as sensor dropouts or new measurements (e.g., EtCO2 upon intubation during transport).

## Labelling adverse events

**Cleaning and preprocessing data.** Data cleaning removed implausible readings: HR > 300 beats per minute (BPM) or <30 BPM, EtCO2 > 15kPa (and airway-derived Respiratory Rate when EtCO2 > 15 kPa), and blood pressure <5mmHg. Readings taken when the monitor was on but not connected to the patient were also removed to ensure each patient had at least 30 minutes of valid data. Finally, data was down-sampled to one-minute averages (mean) to address noise inherent to second-by-second recordings [12].

**Bollinger bands.** To label adverse events during transport, we adapted Bollinger Bands, a technique from financial markets that uses a moving average with upper and lower boundaries set by standard deviations (SD) [13]. This approach has previously been applied in healthcare to identify winter surges in PICU demand [14]. Traditionally defined as ±2 SDs around a 20-day Simple Moving Average (SMA), the parameters (e.g., window size and bandwidth) are flexible depending on the application. By tracking a moving average and its SD, Bollinger Bands monitor stability over time: narrow bands indicate stability with low variability, while wide bands reflect high variability.

A key advantage of Bollinger Bands is that they rely only on the current and preceding data points to update the moving average and calculate standard deviation. Because the method does not require future data ("look-ahead"), it can generate boundaries in real-time, making it well-suited for continuous monitoring scenarios. Each new data point, such as a vital sign measurement, immediately contributes to recalculating the mean and SD, allowing for on-the-fly detection of departures from typical patterns. Compared to alternative time series methods that may require predefined normal ranges, extensive historical data, or assumptions about underlying distributions, Bollinger Bands provide a flexible and patient-specific approach that dynamically adjusts to evolving vital signs without imposing rigid thresholds. This adaptability makes them particularly well-suited for critical care transport, where vital sign baselines vary significantly between patients and evolve over time due to interventions and illness severity.

In the case of detecting patient-specific adverse events, we hypothesised that we could implement Bollinger Bands to track the stability of vital signs over time and use departures from stability to indicate likely occurrence of adverse events.

**Construction of bollinger bands.** In this study, Bands were constructed using an Exponential Moving Average (EMA) instead of SMA as the EMA prioritises recent data while retaining past trends [15]. The EMA was calculated on 1-minute averages, balancing responsiveness to sudden changes without overreacting to transient fluctuations. The span in an EMA determines the degree of weighting applied to recent versus past observations, with shorter spans giving more weight to recent values and longer spans distributing weight more evenly across historical data. Exponentially Weighted Moving Standard Deviations (EWMSTD) were then used to define the upper and lower bands.

To reduce oversensitivity arising from low variability in the data, when a parameter's EWMSTD was within 5% of its current EMA, a fixed boundary of ±5% of the current EMA was applied. Due to its lower variability, oxygen saturation was assigned a stricter fixed boundary of ±2.5% of its current EMA. This adjustment ensures that minor fluctuations, such as a decrease in respiratory rate from 40 to 39 breaths per minute during periods of stability, are not erroneously flagged as clinically significant.

As described below, for both respiratory and cardiovascular vital signs, an EMA with a Span of 15 was chosen. Therefore, at the start of each transport episode, 15 minutes of data was required to "warm-up" and calibrate the bands to the patients baseline. Consequently, adverse events were only labelled and subsequently analysed for periods following this initial 15-minute warm-up phase.

**Adverse respiratory events.** Defined as episodes of respiratory compromise with hypoxaemia (low $SpO_2$), were identified using four parameters: (1) $SpO_2$ (plethysmograph), (2) impedance pneumography derived respiratory rate, (3) airway-derived respiratory rate, (4) EtCO2 [2,16]. Through a process of systematically trialling combinations of span and EWMSTD values and visually inspecting a random subset of the transport episodes values, EMA (Span 15) ± 1.0 EMWSTD, were chosen to construct Bollinger bands for respiratory vital signs. These values effectively captured deviations while minimizing false positives.

Hypoxaemia was identified when $SpO_2$ fell below 94% or its lower Bollinger Band, whichever was lower, ensuring minor changes (e.g., 100–96) did not trigger false positives. To confirm an event, at least one additional parameter (respiratory rate, airway-derived respiratory rate, or end-tidal $CO_2$) had to deviate outside its thresholds simultaneously. Since calculations were performed on vital signs after down-sampling to one-minute averages, a vital sign was required to have an average value outside its bands for at least one minute to qualify as a deviation. Events were validated only if at least five uninterrupted minutes of data were available before and after the deviation to exclude probe signal issues or data interruptions generating false positives.

**Adverse cardiovascular events.** Defined as episodes of significant haemodynamic instability, characterized by deviations in blood pressure (BP) and/or heart rate (HR) [2]. They were identified using three parameters: (1) heart rate (3-lead ECG), (2) non-invasive mean arterial pressure (MAP), and (3) invasive MAP. Through the same process applied to respiratory parameters, EMA (Span 15) ± 1.28 EMWSTD (corresponding to the 90th percentile), were chosen to construct Bollinger bands for cardiovascular vital signs. This accounted for the increased natural variability in HR and BP compared to the respiratory parameters capturing clinically significant deviations while minimizing false positives. Identically to respiratory parameters, we required that at least two parameters deviate outside their respective thresholds to label an event.

An adverse cardiovascular event was identified when both non-invasive and, if available, invasive MAP readings fell outside their thresholds in the same direction (e.g., both high or both low). If only one BP source showed deviation, the event was confirmed if HR also deviated outside its bands whilst simultaneously with at least one BP value. Identically to respiratory parameters, we required at least five uninterrupted minutes of data before and after the deviation.

**Adverse events and minutes of instability.** To quantify respiratory and cardiovascular instability during a transport episode, we separately labelled each averaged minute where respiratory or cardiovascular vital signs deviated outside their dynamically defined stable ranges. The cumulative number of such minutes, calculated separately for respiratory and cardiovascular instability, provides a measure of vital sign instability throughout the transport episode.

An adverse event was defined as a continuous sequence of minutes during which vital signs remain outside the stable range (Fig 3). The first minute of instability marks the start of the adverse event, and the event ends at the last minute before vital signs return to the stable range. The shortest possible duration of an adverse event is one minute, which occurs when vital signs are unstable for only a single minute. Thus, for each transport episode, we measure two key metrics: the total number of adverse events and the cumulative minutes of vital sign instability.

## Validation

To validate labelling, we calculated two parameters per episode: (1) total time of vital sign instability and (2) number of adverse events. We then compared these against three EHR-derived severity markers: (1) respiratory support during transport (2) cardiovascular support during transport (3) 30-day mortality. Respiratory support was defined as any of the following interventions: High Flow Nasal Cannula (HFNC), Continuous Positive Airway Pressure (CPAP), Bilevel Positive Airway Pressure (BiPAP), Invasive Mechanical Ventilation (via endotracheal tube or tracheostomy); patients on supplemental oxygen alone did not meet the criteria, as per recent consensus [17]. Cardiovascular support was defined as the patient receiving at least one vasoactive agent during transport: Adrenaline, Noradrenaline, Dobutamine, Dopamine, Milrinone, Vasopressin.

PLOS Digital Health

## Analysis

All analyses were performed in Python 3.11.6 within the Great Ormond Street Hospital Digital Research Environment (DRE), using packages including NumPy, Pandas, SciPy, Statsmodels, Matplotlib, Seaborn, and Plotly.

## Results

### Cohort description

From July 2016 to May 2021, there were 6,471 transport episodes. Of these, 1,781 had linked transport-monitoring and electronic health record data. After excluding patients older than 18 years and those with fewer than 30 minutes of available monitoring data, 1,519 episodes remained (23.5%; see Fig 1). Most exclusions stemmed from technical issues maintaining a stable connection between the monitor, smartphone, and server. Table 1 compares the demographic, clinical, and transport characteristics of the entire cohort versus the included subset. The distributions are largely similar across the listed variables, indicating that the included subset is representative of the overall dataset. Table 2 details the proportion of transport episodes in which each vital sign was recorded. Not all patients had every parameter measured, as this depended on the clinical situation and ongoing interventions, for example, EtCO2 was recorded only in patients receiving invasive ventilation.

Breakdown of characterists of all transport episodes and included episodes

### Availability of each vital sign parameter in the cohort

### Labelling

Using our personalized, data-driven approach for identifying deviations from individualized normal ranges, we identified at least one adverse respiratory event in 15.7% and one adverse cardiovascular event in 21.5% of 1,519 transport episodes,

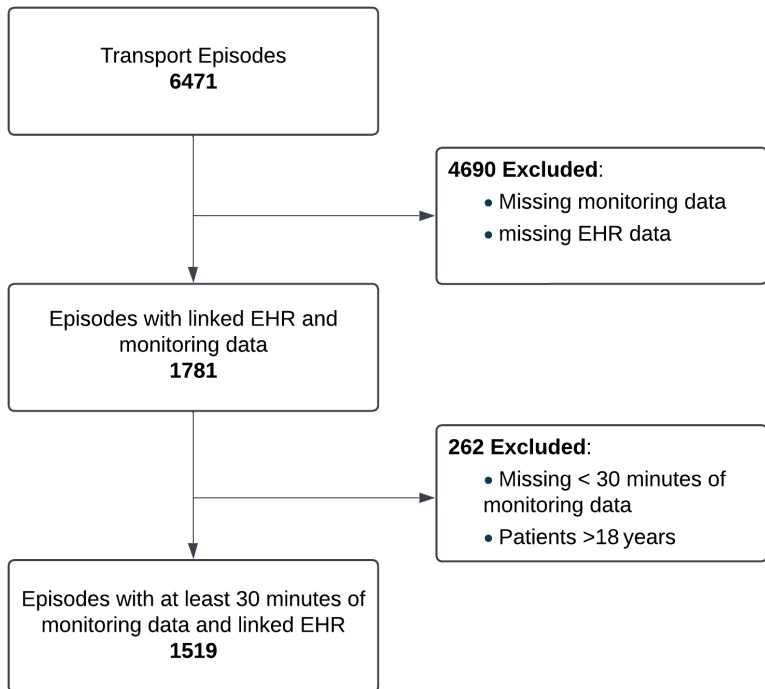

**Fig 1. Flowchart of inclusion criteria.**

**Table 1. Demographic, clinical, and transport characteristics of the study population compared to all transported children during the study period. Respiratory support and cardiovascluar support refer to support received during transport. HFNC = High-Flow Nasal Cannula; CPAP = Continuous Positive Airway Pressure; BIPAP = Bilevel Positive Airway Pressure; ETT = Endotracheal Tube; PIM3 = Paediatric Index of Mortality score 3 [18].**

| Characteristics | All Transport Episodes (n = 6471) | Included Episodes (n = 1519) |
|---|---|---|
| **Age Group Distribution** | | |
| ≤ 1 month (newborn) | 2219 (34.3%) | 568 (37.4%) |
| 1– ≤ 12 months (infant) | 1330 (20.6%) | 296 (19.5%) |
| 1– ≤ 4 years (pre-school child) | 1198 (18.5%) | 264 (17.4%) |
| 4– ≤ 11 years (school child) | 1017 (15.7%) | 239 (15.7%) |
| 11– ≤ 18 years (adolescent) | 678 (10.5%) | 151 (9.9%) |
| > 18 years | 27 (0.4%) | 0 (0%) |
| **Gender Distribution** | | |
| Male | 3603 (55.7%) | 825 (54.3%) |
| Female | 2861 (44.2%) | 692 (45.6%) |
| **Diagnosis Group Distribution** | | |
| Respiratory | 2233 (34.5%) | 498 (32.8%) |
| Cardiovascular | 1385 (21.4%) | 364 (24.0%) |
| Neurological | 921 (14.2%) | 227 (14.9%) |
| Infection | 888 (13.7%) | 202 (13.3%) |
| Gastrointestinal | 368 (5.7%) | 89 (5.9%) |
| Metabolic | 161 (2.5%) | 43 (2.8%) |
| Trauma | 81 (1.3%) | 21 (1.4%) |
| Other | 434 (6.7%) | 75 (4.9%) |
| **PIM3 Risk of Mortality** | | |
| ≤ 1% | 502 (7.8%) | 92 (6.1%) |
| 1– ≤ 3% | 2345 (36.2%) | 510 (33.6%) |
| 3– ≤ 5% | 2060 (31.8%) | 518 (34.1%) |
| 5– ≤ 10% | 1038 (16.0%) | 264 (17.4%) |
| 10– ≤ 15% | 206 (3.2%) | 59 (3.9%) |
| 15– ≤ 30% | 182 (2.8%) | 45 (3.0%) |
| > 30% | 126 (1.9%) | 30 (2.0%) |
| **Respiratory Support** | | |
| Self-ventilating (Room Air) | 1292 (20.0%) | 257 (16.9%) |
| Self-ventilating (supplemental $O_2$) | 141 (2.2%) | 27 (1.8%) |
| Self-ventilating (HFNC) | 250 (3.9%) | 62 (4.1%) |
| Self-ventilating (CPAP) | 278 (4.3%) | 55 (3.6%) |
| Self-ventilating (BIPAP) | 54 (0.8%) | 12 (0.8%) |
| Invasive ventilation (ETT) | 4214 (65.1%) | 1073 (70.6%) |
| Invasive ventilation (Tracheostomy) | 93 (1.4%) | 19 (1.3%) |
| Invasive ventilation (Other airway) | 10 (0.2%) | 3 (0.2%) |
| **Cardiovascular Support** | | |
| Adrenaline | 815 (12.6%) | 199 (13.1%) |
| Dobutamine | 31 (0.5%) | 9 (0.6%) |
| Dopamine | 580 (9.0%) | 137 (9.0%) |
| Milrinone | 58 (0.9%) | 12 (0.8%) |
| Noradrenaline | 507 (7.8%) | 131 (8.6%) |
| Any agent | 2011 (31.1%) | 488 (32.1%) |

*(Continued)*

**Table 1.** (Continued)

| Characteristics | All Transport Episodes (n=6471) | Included Episodes (n=1519) |
|---|---|---|
| **Overall Transport Time, minutes** | | |
| ≤60 | 16 (0.2%) | *0 (0.0%)* |
| 60–120 | 598 (9.2%) | *105 (6.9%)* |
| 120–180 | 1523 (23.5%) | *345 (22.7%)* |
| 180–240 | 2067 (31.9%) | *545 (35.9%)* |
| 240–300 | 1282 (19.8%) | *322 (21.2%)* |
| 300–360 | 556 (8.6%) | *142 (9.3%)* |
| >360 | 261 (4.0%) | *55 (3.6%)* |

**Table 2.** This table details the proportion of transport episodes for which at least one value of a given vital sign was available.

| Vital sign | Data availability (%) |
|---|---|
| $SpO_2$ | 100% |
| Heart rate (3-lead ECG) | 99% |
| Impredance pneumography derived respiratory rate | 96% |
| Non-invasive Blood Pressure | 94% |
| Airway derived respiratory rate | 73% |
| $EtCO_2$ | 73% |
| Invasive Blood Pressure | 36% |

with 6.19% experiencing both. The average duration of an adverse respiratory and cardiovascular events was 91.2 and 82.8 seconds respectively. Examples of respiratory and cardiovascular labels are shown in Figs 2 and 3.

## Respiratory event validation

Respiratory support data was available for 1,508 of 1,519 patients, with 1,224 (81.2%) receiving respiratory support. Kernel density estimates (KDEs) demonstrate that patients requiring respiratory support experience more minutes of respiratory instability (Fig 4A). Logistic regression showed that each additional cumulative minute of respiratory instability increased the odds of receiving support by 20.2% (OR = 1.202; 95% CI: [1.049, 1.378]; p=0.008). For example, five minutes corresponded to a 91.0% probability of support, rising to 96.2% with ten minutes (Fig 5A).

Among 1,519 patients, 30-day mortality data were available for 1,494, with 95 (6.4%) deaths. KDEs demonstrate that patients who died experienced experience more minutes of respiratory instability (Fig 4B). Logistic regression indicated each additional minute of respiratory instability increased the risk of mortality by 19.5% (OR = 1.195; 95% CI: [1.082, 1.320]; p=0.0004). For example, 5 cumulative minutes of instability corresponded to a 12.9% risk of mortality, rising to 26.5% with 10 minutes (Fig 5B).

Re-analysis comparing the number of adverse respiratory events to likelihood of receiving respiratory support and 30-day mortality yielded the same positive relationship (S1 Fig). Each additional event increased the odds of receiving ventilatory support by 30.6% (OR = 1.306; 95% CI: [1.063, 1.604]; p=0.011) and each additional event increased the odds of mortality by 35.5% (OR = 1.355; 95% CI: [1.145, 1.604]; p<0.001) (S2 Fig).

## Cardiovascular event validation

Data were available for all 1,519 patients, with 488 (32.1%) receiving cardiovascular support. Kernel density estimates (KDEs) demonstrate that patients requiring cardiovascular support experience more minutes cardiovascular instability

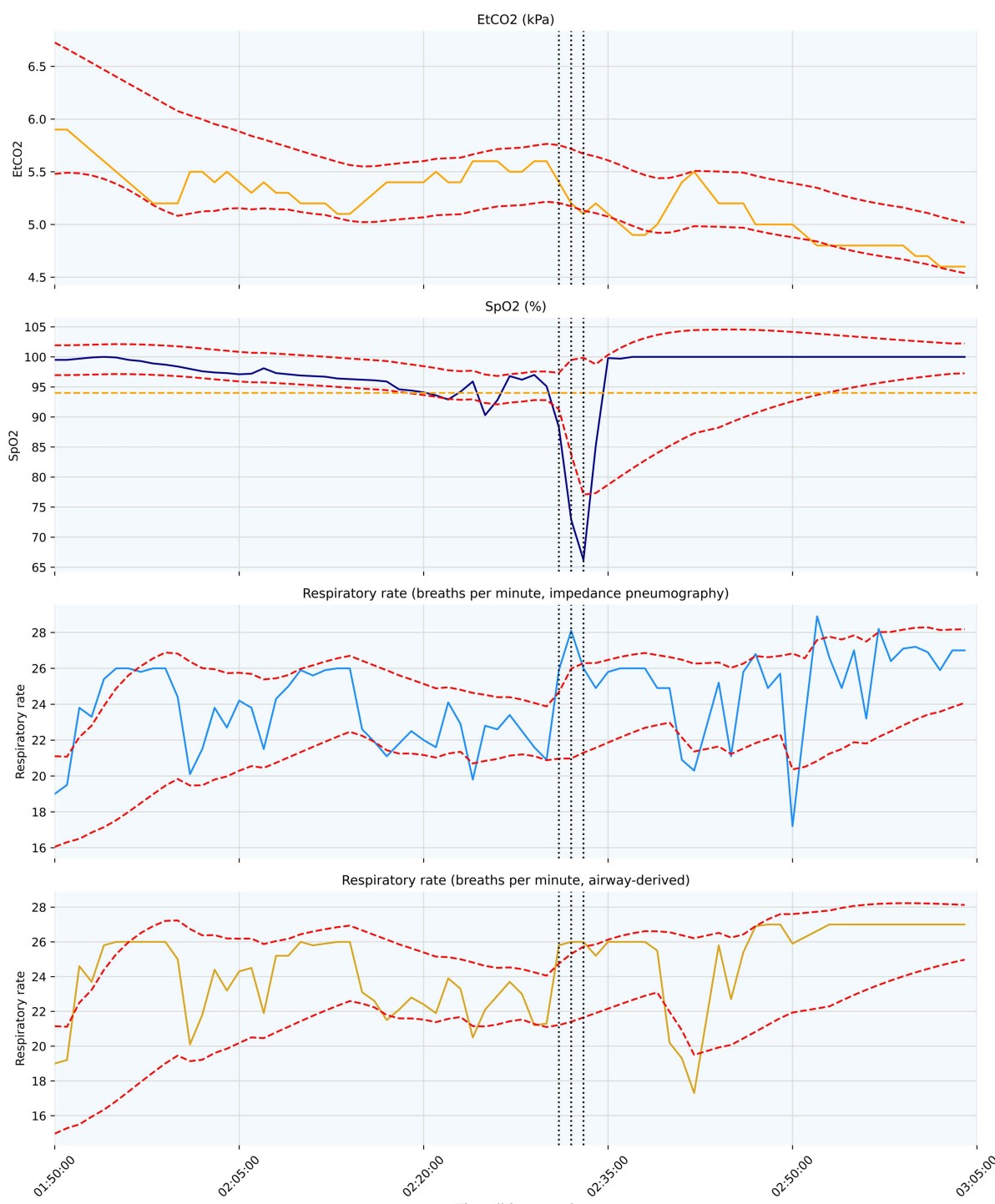

**Fig 2. Variation over time in the four parameters used to identify adverse respiratory events (EtCO$_2$, SpO$_2$, impedance pneumography respiratory rate, and airway respiratory rate).** Dashed red lines indicate parameter boundaries (except SpO$_2$, which has no upper limit). The orange dashed line marks an SpO$_2$ threshold of 94; if the calculated lower bound is > 94, it is capped at 94 to reduce false positives. A profound desaturation, accompanied by a rise in respiratory rate and drop in EtCO$_2$, is labelled with the grey vertical dashed lines, with each line representing one minute of the adverse event.

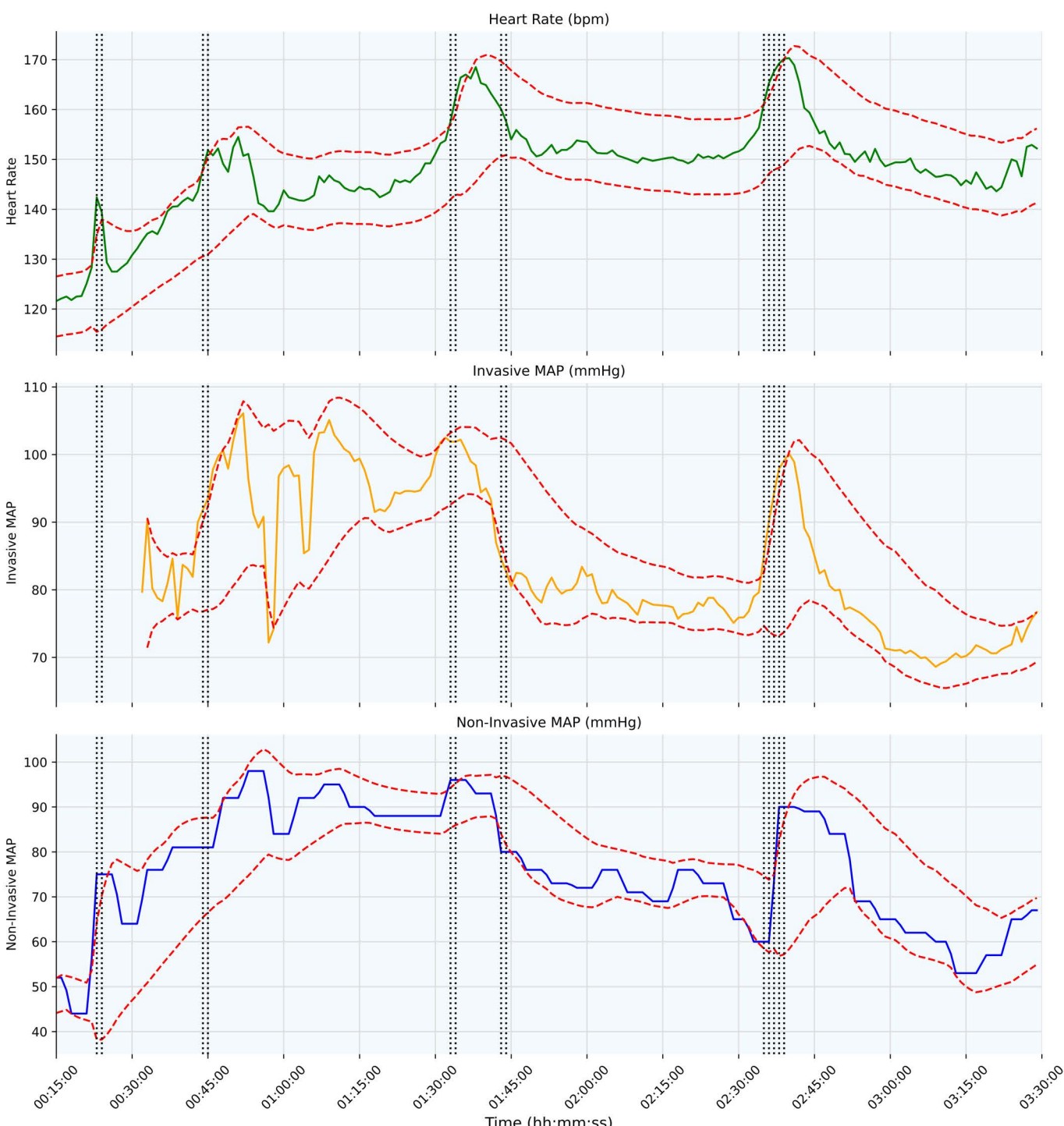

**Fig 3. This figure shows the variation over time in the three parameters used to identify adverse cardiovascular events (HR, invasive MAP, and non-invasive MAP).** Dashed red lines indicate parameter boundaries, and vertical grey dashed lines mark adverse events, characterized by HR spikes and associated BP changes. A total of six discrete adverse cardiovascular events occurs: the shortest lasts one minute at the twelve-minute mark, and the longest, occurring between 02:35 and 02:39, spans five minutes. Overall, these events account for 14 minutes of cardiovascular instability.

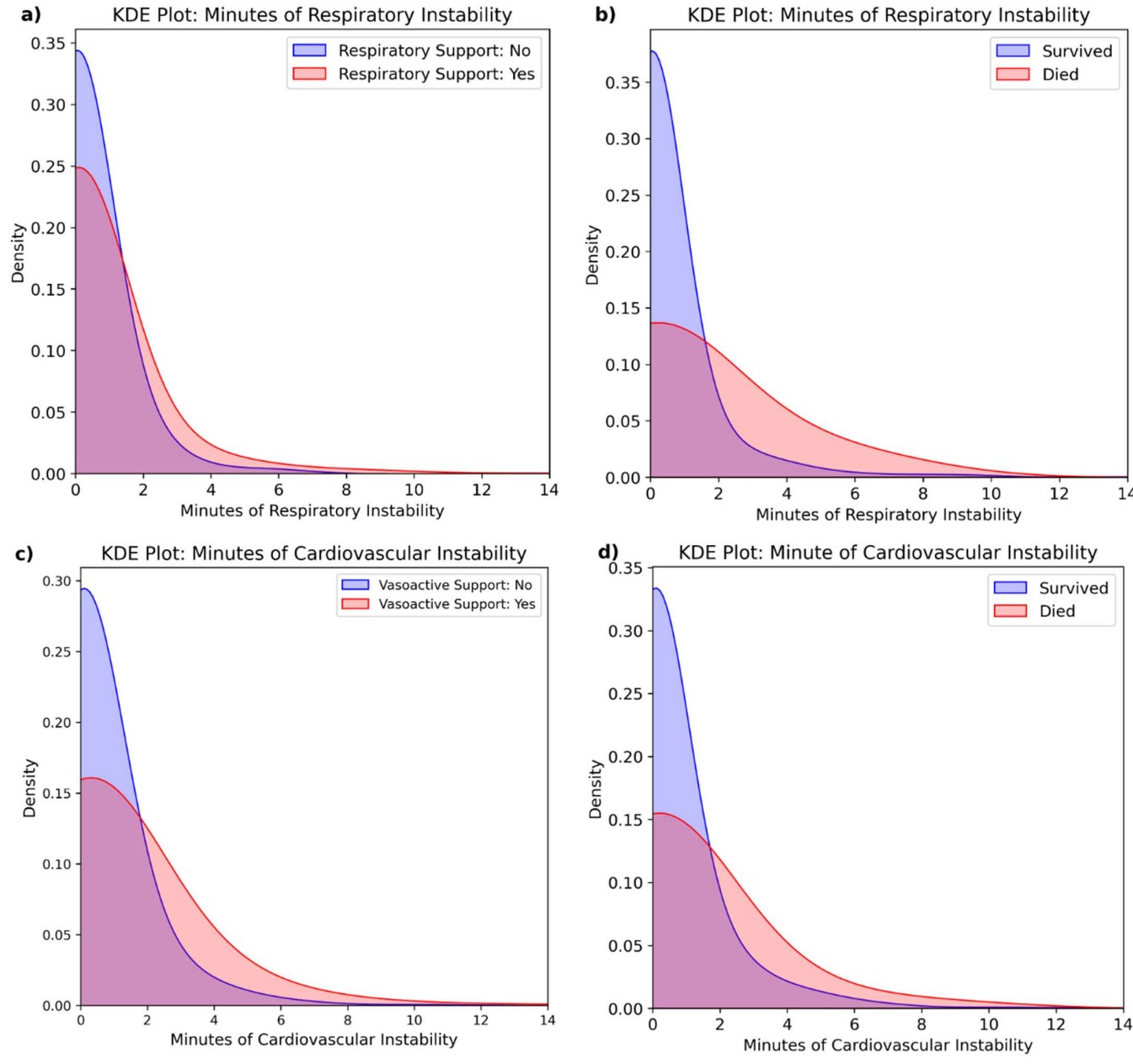

**Fig 4. (a)** KDE plot show that patients with more respiratory instability are more likely to receive respiratory support during transport. **(b)** KDE plot show that patients with more respiratory instability have higher risk of 30-day mortality. **(c)** KDE plot show that patients with more cardiovascular instability are more likely to receive cardiovascular support during transport. **(d)** KDE plot shows that patients with more cardiovascular instability have higher risk of 30-day mortality.

(Fig 4C). Logistic regression demonstrated that each additional cumulative minute of cardiovascular instability raised the odds of requiring support by 27.6% (OR = 1.276; 95% CI: [1.182, 1.378]; p < 0.0001). For example, five minutes corresponded to a 45.0% probability of support, rising to 73.5% with ten minutes (Fig 5C).

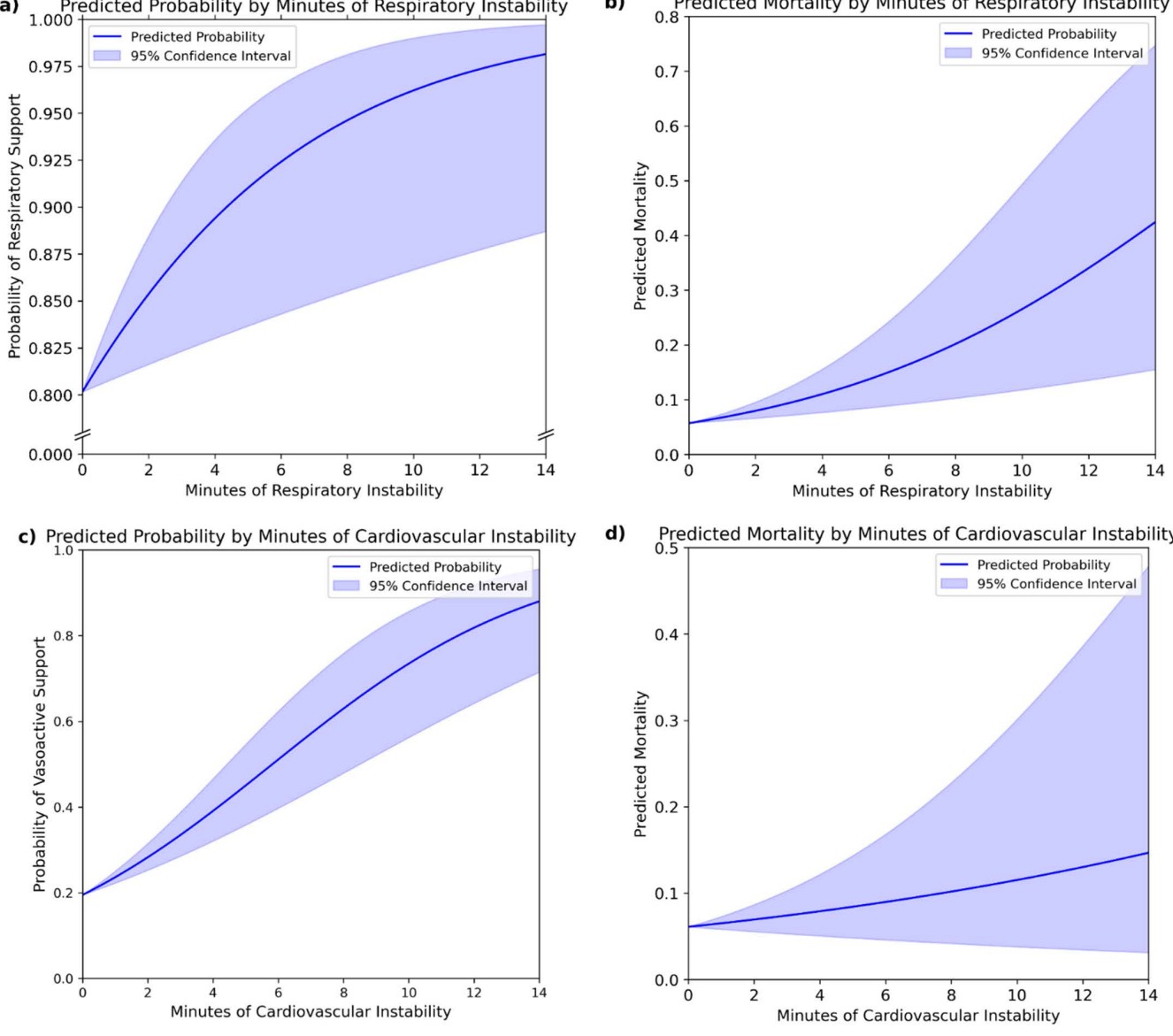

**Fig 5.** (a, b) Logistic regressions indicate that cumulatively longer periods of respiratory instability are associated with increasing probability of receiving respiratory support and 30-day mortality, with blue shaded areas representing 95% confidence intervals. (c, d) Logistic regressions indicate that cumulatively longer periods of cardiovascular instability are associated with increasing probability of receiving cardiovascular support and 30-day mortality, however the trend for 30-day mortality is not statistically significant at the 95% confidence level.

Whilst KDEs suggested that patients who died experienced experience more minutes of respiratory instability (Fig 4D), logistic regression showed no significant relationship: each additional event increased the odds of mortality 7.2% (OR = 1.072; 95% CI: [0.951, 1.208]; p = 0.2545) (Fig 5D).

Re-analysis comparing the number of adverse cardiovascular events to likelihood of receiving cardiovascular support and 30-day mortality yielded the same relationship (S1 Fig). Each additional event increased the odds of receiving cardiovascular support by 34.9% (OR = 1.349; 95% CI: [1.210, 1.503]; p < 0.0001) and each additional event increased the odds of mortality by 7.2% (OR = 1.072; 95% CI: [0.888, 1.293]; p = 0.470) (S2 Fig).

## Discussion

By applying a data-driven labelling method, adapted from financial market analysis to minute-averaged physiological data, we retrospectively identified acute adverse respiratory and cardiovascular events in critically ill children during transport. Our approach dynamically defines each parameter's stable range based on its own historical data and labels an adverse event when multiple parameters deviate. We found that more respiratory events and longer instability were significantly associated with increased receipt of respiratory support and higher 30-day mortality, while more cardiovascular events and longer instability correlated with greater receipt of vasoactive support. These findings suggest that data-driven labelling reliably captures clinically meaningful episodes of acute instability during transport.

Critically ill paediatric patients often do not fit neatly into standard physiological ranges due to demographics, underlying pathology, and ongoing interventions [5,7].For example, heart and respiratory rates depend on age, and some patients, like those with cyanotic heart disease, may have lower baseline oxygen saturations [8,9,19]. Furthermore, specific treatment goals frequently lead to individualized physiological targets that differ from "normal" ranges. For instance, $EtCO_2$ targets can vary significantly (e.g., 4.5-5 kPa for neuroprotection vs. 6–7 kPa for permissive hypercapnia), and SpO2 targets might differ (e.g., > 94% for neuroprotection vs. > 85% for cyanotic heart disease). Similarly, blood pressure and heart rate thresholds are not only age-dependent but also profoundly influenced by vasoactive infusions, used in 32.1% of our cohort. Traditional fixed-threshold alert systems which do not tailor assessments to individual baselines may therefore misclassify or overlook clinically relevant deviations in vital signs. For example, when transport teams first arrive, patients are often at their worst, with vital signs gradually improving during transport [5]. Fixed-threshold alert systems may struggle to distinguish this expected improvement from sudden changes that signal adverse events.

Compared to alternative time series methods that may require predefined normal ranges, extensive historical data, or assumptions about underlying distributions, Bollinger Bands provide a flexible, patient-specific approach that dynamically adjusts to evolving vital signs in real time. By continuously defining stable vital sign ranges at the individual patient level, our approach allows gradual trends to be absorbed within the dynamic stable ranges, resulting in these slow changes not being flagged as adverse events. By focusing on sharp deviations (both abrupt falls and abrupt rises) from an evolving individual baseline, our method helps to differentiate genuine acute deteriorations from expected trends, facilitating timely detection of events that warrant urgent intervention while minimising false positives.

Additionally, defining each parameter's stable range using only prior data prevents reliance on future information, making the system inherently suited for real-time monitoring. This approach ensures that the stability criteria remain free of foresight, preserving the integrity and real-world applicability of the labelling process. Furthermore, down-sampling recorded data from one-second intervals to one-minute intervals, we reduce noise inherent with high frequency recording while preserving the overall trend across clinically relevant timeframes. Transient adverse episodes lasting less than one minute are less likely to be clinically significant [12].

In this cohort of 1,519 interhospital transports, 15.7% experienced adverse respiratory events, 21.5% had adverse cardiovascular events. Singh et al. reported a 12.3% adverse event rate among more than 8,000 interhospital transports, but their lower incidence likely reflects fewer patients requiring mechanical ventilation (11.7%) or exhibiting pre-transport cardiovascular instability (11.4%) compared to our cohort (81.2% and 32.1%, respectively) [20]. Meanwhile, Haydar et al.'s systematic review of intrahospital transports highlights the broad variability of adverse events (respiratory 0.22–69%, cardiovascular 0.66–37.5%), indicating substantial variability across different populations and settings [2]. Overall, our findings align with prior research while emphasizing the importance of tailoring monitoring strategies to each patient's unique physiology.

One limitation of this study is the availability and continuity of patient monitoring data. Despite having 6,471 transports during the study period, only 23.7% (n = 1,519) met the inclusion criterion. Nevertheless, as shown in Table 1, the included subset closely matches the overall transported cohort across major demographic and clinical variables (e.g., age group, diagnosis group), indicating that it remains broadly representative. This underscores the need for

consistent, optimised data collection practices, robust integration with electronic health records, and close collaboration with clinical teams.

Additionally, the emphasis of our methodology on detecting abrupt physiological changes, may overlook insidious, more gradual declines in a patient's condition. This trade-off stems, in part, from the choice of parameters, such as the EMA span and the EWMSTD multipliers. For instance, using a longer EMA span makes the Bollinger Bands adapt more slowly to new data points; while this can increase sensitivity to gradual deteriorations, it also raises the likelihood of falsely flagging small deviations and normal trends as adverse events. Thus, selecting these parameters optimally requires balancing sensitivity to adverse events against the acceptable false positive rate; an equilibrium that ultimately hinges on the methods intended application. In our study it was not possible to rigorously optimize the EMA span and EWMSTD multipliers due to the absence of clinically verified temporal labels for adverse events. Without a gold-standard, we chose an EMA span of 15 and EWMSTD multipliers of ±1.0 (respiratory) and ±1.28 (cardiovascular) by systematically trialling combinations of span and EWMSTD values and visually inspecting a random subset of the transport episodes. Values were chosen to balance sensitivity to adverse events and minimise false positives. However, given the variability in the physiological parameters of critically unwell children, there remains a degree of subjectivity in determining if and when an adverse event occurred.

To validate our data, we explored the possibility of using routinely collected incident notes logged at the end of a transport episode by the clinical team. However, a detailed review of these notes revealed several limitations that undermine their reliability as a gold standard. For instance, while incidents were recorded in 273 of our 1519 episodes, a closer examination of the associated free-text notes showed that many were not relevant to patient deterioration and instead pertained for example to logistical or equipment issues. Furthermore, even among incidents deemed potentially patient-related there was significant ambiguity regarding whether the incident occurred whilst the team was in attendance or if the patient was connected to the transport team's monitor at the time of the incident. This raised significant risk of false negatives and so this dataset was not used in the validation analysis.

Instead, to validate the chosen EMA span and EWMSTD multipliers, we compared labelled events and minutes of instability to surrogate stability markers: respiratory and cardiovascular support during transport. Analysis clearly showed that more respiratory events and longer instability increased the likelihood of receiving respiratory support (Figs 4, 5). Similarly, more cardiovascular events and longer instability were associated with a higher likelihood of receiving cardiovascular support (Figs 4, 5).Thirty-day mortality has been linked to adverse transport events and was therefore used as a second surrogate marker of stability to validate our approach [21]. Our analysis shows a significant association between increasing numbers of adverse respiratory events and mortality but not cardiovascular events (Fig 5). The latter may reflect the low base mortality rate, limiting statistical power, or effective management of cardiovascular stability during transport, mitigating its impact on mortality.

Moreover, the inherent clinical variability of this critically ill paediatric cohort, makes the use of fixed vital sign cutoffs as a direct comparator fundamentally problematic and often clinically inappropriate. As our previous work has demonstrated, vital signs exhibit dynamic trajectories throughout transport, meaning a patient's baseline can gradually shift due to stabilization or interventions [5]. In such a context, an initial fixed cutoff would quickly become unsuitable, leading to either excessive false alarms or a failure to detect actual events as the patient's physiological state evolves. Furthermore, the absence of clinically verified 'ground truth' labels for adverse events in this dataset, presented an additional practical barrier to a rigorous, direct comparison against fixed thresholds. Therefore, a comparison to fixed thresholds would yield limited insights into the true clinical utility of our personalized approach versus what is already recognized as an inherently flawed comparator for this population.

Despite limitations, validation against surrogate stability markers supports our labelling methodology and chosen EMA/EWMSTD settings. The significant link between labelled events and outcomes, such as the need for respiratory or cardiovascular support and mortality, justifies our method as a reliable tool for identifying adverse events in critically ill paediatric transport.

Our computationally lightweight approach not only facilitates retrospective analysis but also has the potential for real-time clinical deployment, enabling clinicians to rapidly identify adverse respiratory and cardiovascular events in real time as they occur. By enabling earlier recognition, this method could improve the clinical response to acute deterioration. Additionally, the time-labelled datasets generated by this method could train machine learning models capable of predicting adverse events before their onset, potentially providing clinicians the opportunity to take preventative measures [22]. Although derived from paediatric transport data, our method is readily adaptable to other high-resolution datasets including adult, paediatric, and neonatal ICU data, since similar high-resolution monitoring is common in these environments. While further prospective validation is required for clinical deployment, the method shows promise for enhancing patient safety and outcomes by augmenting monitoring systems in transport and broader ICU settings.

## Conclusion

We present an automated novel method for labelling adverse respiratory and cardiovascular events during paediatric critical care transports using routinely collected vital signs data. Evaluation of our approach demonstrated that respiratory events and longer instability were significantly associated with increased receipt of respiratory support and higher 30-day mortality, while more cardiovascular events and longer instability correlated with greater receipt of vasoactive support. The approach is readily adaptable to other high-resolution intensive care datasets, for both retrospective labelling as well as automated, real-time identification of adverse events in the clinical setting, offering a foundation for improved monitoring and early intervention in critically ill patients.

## Supporting information

**S1 Fig. (a) KDE plot show that patients with more adverse respiratory events are more likely to receive respiratory support during transport.** (b) KDE plot show that patients with more adverse respiratory events have higher risk of 30-day mortality. (c) KDE plot show that patients with more adverse cardiovascular events are more likely to receive cardiovascular support during transport. (d) KDE plot shows that patients with more adverse cardiovascular events have higher risk of 30-day mortality.
(TIF)

**S2 Fig. (a,b) Logistic regressions indicate that increasing numbers of adverse respiratory events are associated with increasing probability of receiving respiratory support and 30-day mortality, with blue shaded areas representing 95% confidence intervals.** (c) Logistic regressions indicate that increasing numbers of adverse cardiovascular events are associated with increasing probability of receiving cardiovascular support and 30-day mortality, however the trend for 30-day mortality is not statistically significant at the 95% confidence level.
(TIF)

## Acknowledgments

We acknowledge the support from Kinseed for engineering the SwiftCare system used for the real-time extraction and secure server upload of high-frequency vital sign data directly from patient monitors during transport. We also thank,Great Ormond Street Hospital (GOSH) and the GOSH Digital Research Environment team for data curation. All research at Great Ormond Street Hospital NHS Foundation Trust and UCL Great Ormond Street Institute of Child Health is made possible by the NIHR Great Ormond Street Hospital Biomedical Research Centre.

## Author contributions

**Conceptualization:** Milan Kapur, Alexander Brown, Gwyneth Davies, Padmanabhan Ramnarayan.

**Data curation:** Zhiqiang Huo, Philip Knight, Padmanabhan Ramnarayan.

**Formal analysis:** Milan Kapur.

**Funding acquisition:** Milan Kapur.

**Investigation:** Milan Kapur.

**Methodology:** Milan Kapur, Gwyneth Davies, Padmanabhan Ramnarayan.

**Resources:** Milan Kapur.

**Software:** Milan Kapur.

**Supervision:** Kezhi Li, Gwyneth Davies, Padmanabhan Ramnarayan.

**Validation:** Milan Kapur.

**Visualization:** Milan Kapur.

**Writing – original draft:** Milan Kapur.

**Writing – review & editing:** Milan Kapur, Kezhi Li, Alexander Brown, Zhiqiang Huo, Philip Knight, Gwyneth Davies, Padmanabhan Ramnarayan.

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
