## [Decision Letter · Decision Letter 0]

30 Jun 2025

PDIG-D-25-00160Identification of physiological adverse events using continuous vital signs monitoring during paediatric critical care transport: a novel data-driven approachPLOS Digital Health Dear Dr. Kapur, Thank you for submitting your manuscript to PLOS Digital Health. After careful consideration, we feel that it has merit but does not fully meet PLOS Digital Health's publication criteria as it currently stands. Therefore, we invite you to submit a revised version of the manuscript that addresses the points raised during the review process. Please submit your revised manuscript within 30 days Jul 30 2025 11:59PM. If you will need more time than this to complete your revisions, please reply to this message or contact the journal office at digitalhealth@plos.org. Please include the following items when submitting your revised manuscript:* A rebuttal letter that responds to each point raised by the editor and reviewer(s). You should upload this letter as a separate file labeled 'Response to Reviewers '. This file does not need to include responses to any formatting updates and technical items listed in the 'Journal Requirements' section below.* A marked-up copy of your manuscript that highlights changes made to the original version. You should upload this as a separate file labeled 'Revised Manuscript with Track Changes '.* An unmarked version of your revised paper without tracked changes. You should upload this as a separate file labeled 'Manuscript '. If you would like to make changes to your financial disclosure, competing interests statement, or data availability statement, please make these updates within the submission form at the time of resubmission. Guidelines for resubmitting your figure files are available below the reviewer comments at the end of this letter. We look forward to receiving your revised manuscript. Kind regards, Peter H Charlton, MEng, PhDSection EditorPLOS Digital Health Leo Anthony CeliEditor-in-ChiefPLOS Digital Healthorcid.org/0000-0001-6712-6626  **Journal Requirements:** **Additional Editor Comments (if provided):** Thank you for this submission. In addition to the reviewers' comments, please ensure you are familiar with the PLOS policies on:

- Data availability: https://journals.plos.org/digitalhealth/s/data-availability

- Sharing code: https://journals.plos.org/digitalhealth/s/materials-software-and-code-sharing

I understand that checks would be made by the journal prior to any potential publication, so thought it might be helpful to highlight these at this stage. **Reviewers' Comments:** Reviewer's Responses to Questions

**Comments to the Author**

1. Does this manuscript meet PLOS Digital Health’s publication criteria ? Is the manuscript technically sound, and do the data support the conclusions? The manuscript must describe methodologically and ethically rigorous research with conclusions that are appropriately drawn based on the data presented.

Reviewer #1: Yes

Reviewer #2: Yes

2. Has the statistical analysis been performed appropriately and rigorously?

Reviewer #1: I don't know

Reviewer #2: Yes

3. Have the authors made all data underlying the findings in their manuscript fully available (please refer to the Data Availability Statement at the start of the manuscript PDF file)?

Reviewer #1: Yes

Reviewer #2: No

4. Is the manuscript presented in an intelligible fashion and written in standard English?

Reviewer #1: Yes

Reviewer #2: Yes

5. Review Comments to the Author

Reviewer #1: Whilst I am not an expert on the statistical processes used in this paper I found it an accessible read and a coherent manuscript.

This appears to be a novel application of a process from an non-healthcare industry which will have a practical application.

It seems reasonable that this paper will create an evidence base for further research in this field.

I'd be interested in to know if patients being transported, by definition having been stabilised to some extent, are different from an vital signs perspective from patients conveyed to hospital as an emergency presentation (i.e from home). This may cause this application of this approach to be relevant only to transfers (but I suspect it does have wider application).

I did also wonder whether gradual trends of increase or decrease where associated with specific outcomes and wonder if this could be measured?

Reviewer #2: Thank you for this interesting study on detecting physiological adverse events from vital signs data during pediatric transport. I thought the associations between the detected events and short- and long-term outcomes were particularly interesting. However, I note that there was no comparator approach (such as fixed vital-sign cutoffs) against which to compare the novel approach which calculates personalised and time-varying normal ranges. Therefore, it was difficult to evaluate the potential benefit of the proposed approach beyond existing approaches.

Major Comments:

- Introduction: Currently, the aim of the study isn't entirely clear from the Introduction: "we propose and validate a novel data-driven approach for automatically labelling patient-related adverse events, specifically respiratory and cardiovascular, using routinely collected vital signs data from the transport of critically unwell children." Could you also provide some insight into the way in which the proposed approach will be assessed, e.g. investigating the association between detected physiological adverse events and outcomes (namely treatment and mortality)?

- Conclusion: I think the conclusion should be rephrased in part to ensure it is supported by the results. For instance:

(i) "This approach reliably identified clinically significant events" - I don't think the results demonstrate that the detected physiological adverse events were clinically significant events. Rather, I think they demonstrated that "respiratory events and longer instability were significantly associated with increased receipt of respiratory support and higher 30-day mortality, while more cardiovascular events and longer instability correlated with greater receipt of vasoactive support". In my view, this is a little difference to identifying "clinically significant events", because "clinically significant events" suggests to me an event that requires a response, rather than a change that is associated with a future outcome. Perhaps I have misinterpreted this - in which case perhaps clarification would be helpful.

(ii) "The resulting labelled datasets pave the way for predictive AI models" - I don't think there is evidence in the results to support this statement (or at least in the way it is currently phrased), and I would suggest either omitting or at least re-phrasing it.

Minor Comments:

- Methods > Data Sources: Please could you state what sensing modalities were used to measure each vital sign? For instance, the manuscript currently states "vital signs, including heart rate, respiratory rate, blood pressure, oxygen saturation (SpO2), and end-tidal carbon dioxide (EtCO2)". However, at this point the reader is not aware of whether BP was invasive or non-invasive; whether heart rate was derived from ECG or pulse oximetry (or whichever is available); and the sensor used to obtain respiratory rate (e.g. impedance pneumography from ECG leads, or EtCO2 monitoring).

- Methods > Data Sources: Similarly, this same sentence states "high-resolution (one data point per second) vital signs, including ... blood pressure". But surely when non-invasive BP is measured it doesn't provide new measurements each second. Perhaps cuff-based measurements are held at their previous value until the next cuff measurement? Please clarify if and how non-invasive BP measurements are measured at one data point per second.

- Methods > Inclusion Criteria: "had at least 30 minutes of vital sign data". Please state somewhere in the manuscript (not necessarily here), how long it took to learn a patient-specific baseline (e.g. 15 mins, from EMA span of 15), and also, which period of time predictions of physiological adverse events were made on. For instance, if it does take 15 minutes to establish a baseline, then perhaps predictions are only made after the first 15 minutes of recording?

- Methods > Adverse Respiratory Events: "identified using four parameters: (1) SpO₂ (plethysmograph), (2) plethysmograph-derived respiratory rate". This gave me the impression that both SpO2 and respiratory rate were obtained from the same sensor (called a plethysmograph here, which presumably refers to either photoplethysmography used in pulse oximeters, or a respiratory belt using inductance plethysmography or similar). Please make it clear (as per my earlier comment) which sensor(s) are used for which vital signs.

- Results > Table 1: Please define "PIM3" in the caption

- Results > Labelling: "Using this technique, we identified at least one adverse respiratory event..." Please state the technique, as I don't think it's entirely clear which technique "this technique" refers to in this sentence.

- Results > Figure 2: appears pixelated in my PDF version. Please ensure the figures are clear.

- Acknowledgements: Please provide further details of the support from Kinseed.

6. PLOS authors have the option to publish the peer review history of their article (what does this mean? ). If published, this will include your full peer review and any attached files.

**Do you want your identity to be public for this peer review?** For information about this choice, including consent withdrawal, please see our Privacy Policy .

Reviewer #1: **Yes: ** Damian Roland

Reviewer #2: **Yes: ** Peter H Charlton

---

## [Editor Report · Decision Letter 1]

2 Sep 2025

Identification of physiological adverse events using continuous vital signs monitoring during paediatric critical care transport: a novel data-driven approach

PDIG-D-25-00160R1

Dear Dr. Kapur,

We're pleased to inform you that your manuscript has been judged scientifically suitable for publication and will be formally accepted for publication once it meets all outstanding technical requirements.

Within one week, you'll receive an e-mail detailing the required amendments. When these have been addressed, you'll receive a formal acceptance letter and your manuscript will be scheduled for publication.

An invoice for payment will follow shortly after the formal acceptance. To ensure an efficient process, please log into Editorial Manager at https://www.editorialmanager.com/pdig/ click the 'Update My Information' link at the top of the page, and double check that your user information is up-to-date. For billing related questions, please contact billing support at https://plos.my.site.com/s/.

Kind regards,

Peter H Charlton, MEng, PhD

Section Editor

PLOS Digital Health

Additional Editor Comments (optional):

Thank you for a fascinating manuscript.